# Tuberculosis care provided by private practitioners in an urban setting in Indonesia: Findings from a standardized patient study

Bony Wiem Lestari[1,2,3], Panji F. Hadisoemarto[1,2,4]*, Nur Afifah[1],
Susan McAllister[4], Deny Fattah[1], Argita D. Salindri[1], Reinout van Crevel[3,5],
Megan Murray[6], Philip C. Hill[4], Bachti Alisjahbana[1,7]

1 Tuberculosis Working Group, Research Center for Care and Control of Infectious Diseases, Universitas Padjadjaran, Bandung, Indonesia, 2 Department of Public Health, Faculty of Medicine Universitas Padjadjaran, Bandung, Indonesia, 3 Department of Internal Medicine and Radboud Center for Infectious Diseases, Radboud University Medical Center, Nijmegen, the Netherlands, 4 Centre for International Health, Division of Health Sciences, University of Otago, Dunedin, New Zealand, 5 Centre for Tropical Medicine and Global Health, Nuffield Department of Medicine, University of Oxford, Oxford, United Kingdom, 6 Department of Global Health and Social Medicine, Harvard Medical School, Boston, Massachusetts, United States of America, 7 Department of Internal Medicine, Faculty of Medicine Universitas Padjadjaran, Dr Hasan Sadikin General Hospital, Bandung, Indonesia

☯ These authors contributed equally to this work.
* panji.fortuna@unpad.ac.id

**Data Availability Statement:** The data that support the findings of this study are openly available in

## Abstract

In Indonesia, government-owned Community Health Centers (CHCs) spearhead tuberculosis (TB) care at the primary level, but a substantial proportion of individuals with pulmonary TB also seek care from Private Practitioners (PPs). However, little is known about PPs' practice in managing patients with TB-associated symptoms. To avoid bias associated with self-administered surveys, we used standardized patients (SPs) to evaluate PPs' adherence to the national TB guidelines. Four clinical scenarios of individuals presenting complaints suggestive of TB, accompanied by different sputum smear results or TB treatment histories were developed. We assigned 12 trained SPs to PPs practicing in 30 CHC catchment areas in Bandung city, Indonesia. For comparison, two scenarios were also presented to the CHCs. A total of 341 successful SP visits were made to 225 private general practitioners (GPs), 29 private specialists, and 30 CHCs. When laboratory results were not available, adherence to the recommended course of action, i.e., sputum examination, was low among private GPs (31%) and private specialists (20%), while it was requested in 87% of visits to the CHCs. PPs preferred chest X-ray (CXR) in all scenarios, with requests made in 66% of visits to private GPs and 84% of visits to private specialists (vs. 8% CHCs). Prescriptions of incorrect TB drug regimens were reported from 7% and 13% of visits to private GPs and specialists, respectively, versus none of the CHCs. Indonesian PPs have a clear preference for CXR over microbiological testing for triaging presumptive TB patients, and inappropriate prescription of TB drugs is not uncommon. These findings warrant actions to increase awareness among PPs about the importance of microbiological testing and of administering appropriate TB drug regimens. SP studies can be used to assess the impact of these interventions on providers' adherence to guidelines.

OPENICPSR at https://www.openicpsr.org/openicpsr/, reference number 197521.

**Funding:** The study was funded by Partnership for Enhanced Engagement in Research (PEER) grant under Prime Agreement Number AID-OAA-A-11-00012 by National Academy of Sciences (NAS). BWL received a scholarship from the Indonesian Endowment Fund for Education (LPDP). PFH acquired scholarship from the University of Otago, New Zealand. The funder had no role in study design, data collection and analysis, decision to publish or preparation of the manuscript.

**Competing interests:** The authors have declared that no competing interests exist.

## Introduction

Indonesia has the second highest tuberculosis (TB) burden worldwide, with approximately one million incident cases annually [1]. The National TB Program (NTP) provides free TB-related services in the public sector, including diagnostics and treatment, but fragmented healthcare delivery by providers in the public and private sectors presents a challenge to TB control. A survey of 414 pulmonary TB patients undergoing treatment in Bandung, West Java, showed that 75% of these patients first sought care from informal providers or private practitioners (PPs) and about 40% started treatment in the private sector [2]. A national-level analysis of TB patient pathways reported similar findings and further identified that close to 20% of all notified TB cases initially sought care in private primary care facilities, including PPs. The probability of TB patients initially visiting private primary care facilities with access to diagnostic facilities was only 2% and, in addition, only 16% of these providers were linked to the NTP network for TB diagnostics and treatment [3]. Consequently, patients who access TB care outside of the NTP network may receive sub-optimal care.

As few PPs are linked to the NTP network, little is known about their diagnostic and treatment practices for presumptive and confirmed TB cases. Over half of the PPs surveyed in a multi-city study in Indonesia reported having used sputum smear microscopy for diagnosing TB [4]. However, as previous surveys relied on self-reported data gathered through interviews, results may be subject to the Hawthorne effect or social desirability bias. This could be improved by using standardized patients (SPs), as has been done in other settings [5–8].

In this study, we aimed to assess the quality of TB case management delivered by PPs in Bandung, Indonesia, by using SPs. To evaluate PPs' behaviors in managing adult pulmonary TB cases under different clinical scenarios, and to compare the PPs' clinical decisions to that of providers in the public sector, we developed SP scenarios appropriate for use in Indonesian settings.

## Methods

### Study setting

The study was conducted in Bandung city from July 2018 to April 2019. At that time, Bandung had a population of approximately 2.4 million people, a TB case notification rate of 339/100,000 population (approximately 12,000 cases/year), and a TB treatment success rate of 83% for drug-sensitive TB [9]. Similar to the rest of Indonesia, sputum smear microscopy, rapid sputum molecular test using Xpert MTB/Rif (Xpert), and TB drugs are provided free of charge by public healthcare providers, namely at community health centers (CHCs, or locally known as *puskesmas*) and public hospitals, as well as private hospitals that are a part of the NTP network. The TB program in the city is supported by a network that connects 73 CHCs, one lung clinic, one lung hospital, 16 secondary-level hospitals, four prison clinics, and one tertiary-level hospital. This SP study was part of a group of studies into services delivered for TB in the private sector, examining the role of PPs in TB care in Indonesia [10]. When this study was undertaken, the vast majority of primary-level private healthcare facilities, including PPs, were not a part of the NTP network.

During the course of the study period, the NTP activities adhered to the national guidelines for TB control issued by the Indonesian Ministry of Health in 2016 [11]. According to these guidelines, sputum smear examination is recommended for all cases presenting with main pulmonary TB symptoms, i.e., a productive cough lasting for at least two weeks with or without other accompanying symptoms. Subsequently, individuals with a negative sputum smear can be followed up with a chest X-ray (CXR) examination or prescribed with broad-spectrum

antibiotics. Otherwise, bacteriologically confirmed TB cases can undergo TB treatment. At the time of the study, sputum Xpert examination was prioritized for individuals with a high probability of drug-resistant TB, such as those with a previous history of undergoing TB treatment.

## Study design

The present study used SPs, sometimes referred to as simulated patients or more generally as mystery shoppers, a structured, concealed participant observation method commonly used in business and management science [12]. The use of SPs ensures that the observed behaviors occur naturally, overcoming potential weaknesses of surveys, qualitative interviewing, or unconcealed participant observation [13]. Even though concealment, and therefore lack of consent, may raise ethical issues, the use of SPs can be justified when considering the value of the knowledge gained, especially when the method can produce more scientifically valid results compared to its alternatives and when the risk, in this case the breach of confidentiality, can be minimized or eliminated [14].

## Study participants

The study location was 30 randomly selected CHC areas, each covering 1–6 village-equivalent (Indonesian: *kelurahan*) administrative areas, in Bandung. From a mapping survey to identify actively practicing PPs, conducted between August 2017 and April 2018 in the area, we identified a total of 1106 PPs, comprising of 782 General Practitioners (GPs), 32 internists or pulmonologists, and 292 other specialists. PPs were included in the sampling frame if they were qualified as a GP, an internist, or a pulmonologist, and provided general outpatient care either in single provider or multi-provider clinics in the study area. We excluded PPs in specialized practices who were not providing services to adult patients with general symptoms (e.g., skin clinics). In the mapping survey, we also interviewed the PPs to collect basic information on demographic characteristics and TB-related services. This information, however, was not collected from GPs practicing in the CHCs.

## Sample size

A study in India reported that about 30% of PPs treated TB cases correctly [7]. Assuming this number and to obtain an absolute precision of ± 5% at 95% confidence interval (CI), we estimated that 256 PPs would be needed for this study. We added 25% of the intended sample size to account for possible unsuccessful visits. Considering the small number of eligible specialists in the study area, we applied the sample size calculation only to private GPs. Hence, a total of 320 GPs were selected by stratified random sampling based on CHC area, gender, and the reported number of TB cases managed in the past three months. We included all (n = 32) identified internists/pulmonologists practicing in the selected CHC areas, as well as the 30 CHCs where SPs would be seen by GPs.

## Clinical scenarios

We adapted SP scenarios from the study conducted in India to fit into the Indonesian context [7]. Specifically, we designed four scenarios to reflect different 'decision nodes' in the TB diagnosis and treatment flowchart as written in the NTP 2016 guideline (Fig 1) [11]. Each scenario represented an adult with classic pulmonary TB-associated symptoms, i.e., a chief complaint of having a productive cough for one month, accompanied with different information on TB diagnostics or treatment history, as follows: scenario (A) with no sputum test result and no previous history of TB; (B) with a negative sputum smear result and no previous history of TB;

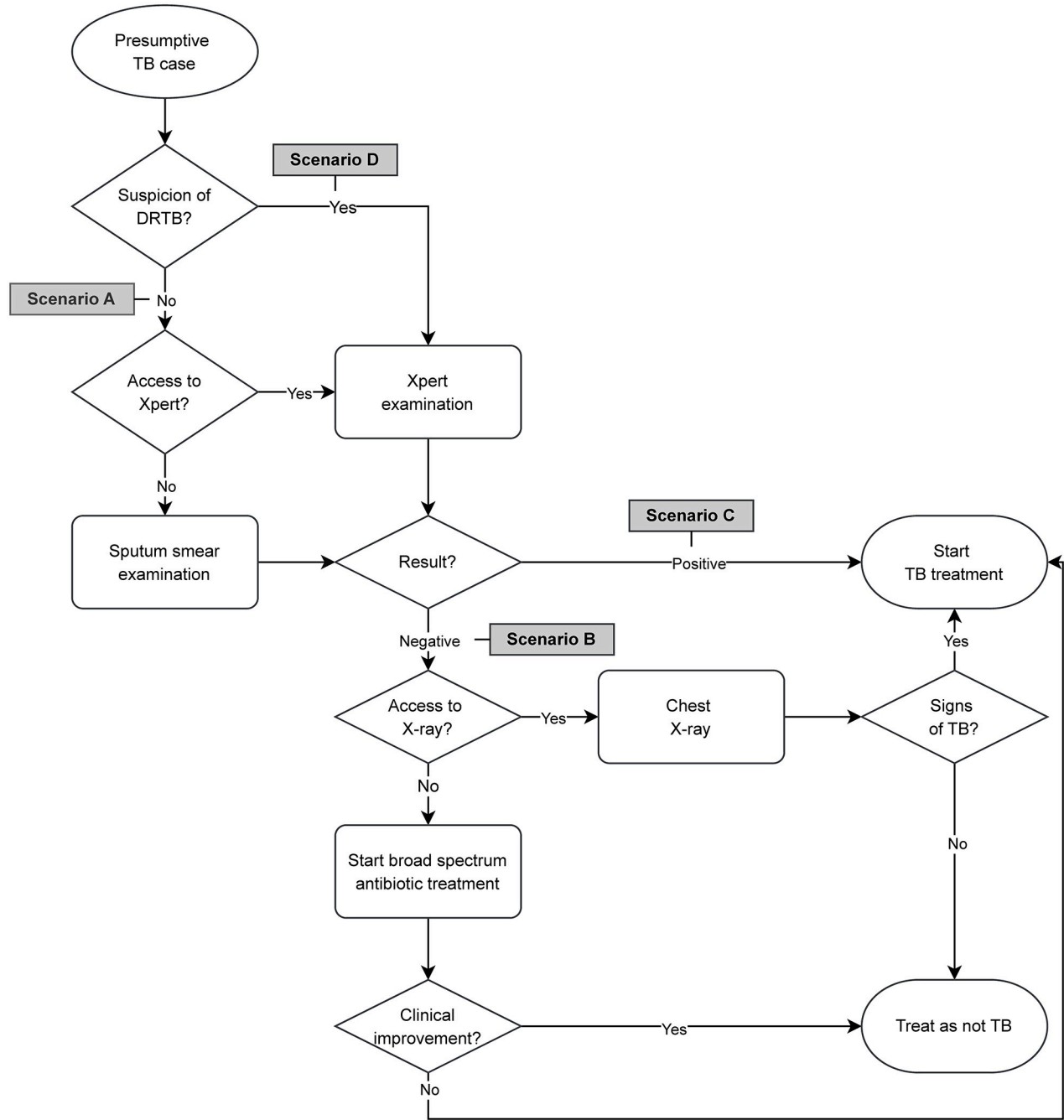

**Fig 1. Presumptive pulmonary tuberculosis (TB) case management flowchart according to the Indonesian National Tuberculosis Program guideline.** Four standardized patient scenarios were developed to represent the "decision nodes" associated with certain patient characteristics and access to a number of diagnostic modalities. Abbreviation: DRTB: Drug-resistant tuberculosis.

(C) with a positive sputum smear result and no previous history of TB; (D) with no sputum test result and a history of recent default from TB treatment. The first draft of scenarios (written by BA and PFH) were discussed in a series of focus group discussions with local TB experts, including pulmonologists and GPs from the local referral hospitals, and NTP staff at

the city health office. Each scenario had a standardized opening statement and scripted presentation that would facilitate the practitioners towards a TB diagnosis and consider the appropriate case management that, unlike the national guidelines, could include requesting sputum testing or a CXR initially, treatment initiation, or a referral to a qualified provider or health care facilities. Sputum smear result documents for scenarios B and C were kindly issued by the city health office's clinical laboratory every two weeks.

Twelve professional actors (9 males, 3 females, age 22–46 years), with previous experience as SPs in medical school examinations, were recruited to present the four clinical scenarios. To maintain the consistency of patient-provider interactions, we developed standard operating procedures (SOPs) for presenting the clinical scenarios. They were trained over two sessions to follow the SOPs and consistently present the cases to investigators with clinical experience with TB patients (BWL, BA). SPs were trained to provide only chief complaints and scenario-relevant laboratory results at the beginning of the consultation; other supporting information were not provided unless prompted by the physicians. In addition, SPs were trained to avoid invasive procedures to be performed during their interactions with the physicians.

## Data collection

Each of the four SP scenarios was randomly assigned to 80 private GPs. Since the number of internists/pulmonologists in the study area was small, we randomly assigned two scenarios to each of these providers, presented by different SPs, such that each scenario was assigned to 16 different specialists. Based on our consultation with the local NTP, only scenarios A and D were assigned to CHCs to avoid the initiation of treatment to SPs presenting sputum smear results that would lead to recording and reporting of the SPs by CHCs to the NTP.

After each visit, SPs completed an exit questionnaire collecting information on the time and length of visit, the number of patients in the waiting room, history taking, physical examination, suggested diagnostic workup, referral for treatment, and any medical costs. We validated the SOP and exit questionnaire on 40 SP visits, for which investigators (DF, NA) compared exit questionnaire responses with voice recordings taken during the patient-provider interaction. Weekly evaluations were held as a quality control measure and to distribute assignments for the next week's visits.

## Data analysis

Adherence to the NTP guidelines and other categorical variables were presented as percentages and 95% confidence intervals (CI). Continuous variables such as age, number of TB cases diagnosed or treated in the past three months, and the duration of patient-provider interaction were presented as a median and its interquartile range (IQR). Duplicate visits to physicians were excluded from analysis. Analyses were performed in SAS Studio (SAS Institute Inc.). CIs for percentages were calculated as exact CI for binomial proportion using PROC FREQ, multiplied by 100. Graphical presentations were produced using ggplot function in R Statistical Software (v4.2.1; R Core Team 2022).

We considered"adherence with the NTP guidelines" to be a recommendation for sputum smear or Xpert examination (Scenario A), a request for CXR (Scenario B), a correct prescription of anti-TB drugs (Scenario C), and a recommendation for sputum test using Xpert (Scenario D), requested either in writing or verbally. In addition, we also considered alternative clinical algorithms where the use of CXR was appropriate as a diagnostic aid for presumptive TB cases in all the scenarios except for Scenario C (Table 1) [15].

**Table 1. Description of scenarios based on the Indonesian National Tuberculosis (TB) Program guidelines for TB control and the World Health Organization (WHO) recommendations for the use of radiography in TB detection.**

| Clinical Scenario | Presentation of standardized patient | Expected management* | Role of chest x-Ray as TB diagnostic aid** |
|---|---|---|---|
| A | Classic case of presumptive TB; no sputum smear result; no previous history of TB | Recommendation for sputum smear microscopy/ sputum Xpert | Yes |
| B | Classic case of presumptive TB; a recent negative sputum smear result; no previous history of TB | Recommendation for chest x-ray or prescription of non-TB antibiotics | Yes |
| C | Classic case of presumptive TB; a recent positive sputum smear result; no previous history of TB | Initiation of treatment with standard first line anti-TB medication (HRZE) | No |
| D | Classic case of presumptive TB; no sputum smear result; a history of incomplete TB treatment | Recommendation for sputum Xpert, or bacterial culture with drug susceptibility test | Yes |

* MoH RoI, 2016[11];

** WHO, 2016[15]

## Ethics statement

Informed consent was not sought from PPs in this study, to avoid bias. Ethical approval, including a waiver for informed consent from research participants, was obtained from the Health Research Ethics Committee of Universitas Padjadjaran (number 687/UN6.C.10/PN/2017).

## Results

### Study participants and SP visit success

A total of 628 GPs and 32 specialists practicing in the 30 CHC catchment areas were eligible for participation. The SPs successfully visited 254 of the 320 randomly selected PPs (Table 2). About half of the visited private GPs were females, whereas the majority (72%) of the specialists were males. Specialists also tended to be older than GPs. Most of the PPs practiced in healthcare facilities with more than one practicing physician, close to a quarter practiced in a facility equipped with laboratory facilities, and about 7% practiced in a facility equipped with an X-ray machine. Eight (4%) of the private GP practices also offered sputum smear microscopy testing. Over a quarter of the private GPs provided services in clinics that accepted national health insurance patients, compared to none among the specialists. PPs diagnosed a median of 2 (IQR 0–3) TB patients in the past three months, but nearly half (48%) had not diagnosed any TB patients in the past three months.

Between July 2018 and April 2019, the SPs successfully made 341 out of 444 (77%) planned PP visits (Fig 2). The visit success was lower to private GPs (71%) compared to that of specialists (89%), with Scenario A visits to private GPs having the lowest success (65%) and Scenario B visits to specialists having the highest success. On one occasion the SP decided to present the simpler Scenario A in lieu of Scenario D. The SP made this decision because he knew the physician had recognized him as one of the actors used for medical school examinations. Otherwise, SPs did not report any suspicion from physicians about the purpose of visits. All visits to CHCs were successful.

Of the 103 unsuccessful visits, most (83%) were due to physicians being temporarily (e.g., on leave) or permanently (e.g., retired, the clinic was permanently closed) not practicing. In a few cases (n = 6), SPs presented an incorrect scenario, either unintentionally (e.g., showing a negative sputum smear result instead of a positive one) or intentionally.

**Table 2. Socio-demographic characteristics and tuberculosis-related management practices of visited private practitioners and the community health centers.**

| Characteristic | General Practitioners n = 225, n (%) | Specialists n = 29, n (%) | Community Health Centers N = 30, n (%) |
|---|---|---|---|
| Male | 110 (51.1) | 21 (72.4) | N/A |
| Median age in years (IQR) | 41 (32–54) | 71 (52–74) | N/A |
| Missing | 51 | 18 | |
| Type of clinic | | | |
| *Multi-provider* | 186 (82.7) | 23 (79.3) | 0 (0.0) |
| *Single-provider* | 39 (17.3) | 6 (20.7) | 30 (100.0) |
| Supporting facilities at clinics | | | |
| X-ray | 16 (7.1) | 2 (6.9) | 0 (0.0) |
| Clinical laboratory | 54 (24.0) | 7 (24.1) | 30 (100.0) |
| Sputum examination | 8 (3.6) | 0 (0.0) | 26 (86.7) |
| Pharmacy | 155 (68.9) | 22 (75.9) | 30 (100.0) |
| Registered with the national health insurance networks | 60 (26.7) | 0 (0.0) | 30 (100.0) |
| Number of TB diagnoses in the past 3 months | | | |
| 0 | 99 (44.0) | 22 (75.9) | 0 (0.0) |
| 1–4 | 92 (40.9) | 3 (10.3) | 9 (30.0) |
| 5 or more | 34 (15.1) | 4 (13.8) | 21 (70.0) |

Abbreviations: IQR: interquartile range; N/A: not available; TB: tuberculosis

## Adherence to TB guidelines

All visits considered, adherence to NTP guidelines for TB management were 52% (95% CI: 45.7–59.1%), 48% (95% CI: 34.7–61.9%), and 45% (95%CI: 32.1–58.4%) for private GPs, specialists, and CHCs, respectively. When CXR was considered as an appropriate diagnostic pathway for Scenarios A, B, and D, percentages of guideline adherence were 78% (95% CI: 72.3–83.4%), 84% (95% CI: 71.7–92.4%), and 50% (95% CI: 32.1–58.4%) for private GPs, specialists, and CHCs, respectively.

For Scenario A, where sputum examination was the recommended course of action, adherence to NTP guidelines among private GPs (number of visits, n = 52) and specialists (n = 15) were lower compared to CHCs (n = 30) (31% vs. 20% vs. 87% for private GPs, specialists, and CHCs, respectively) (Fig 3A and S1 Table). If CXR examination was considered as an acceptable diagnostic aid, guideline adherence increased to 67% for private GPs and 87% for specialists. CXR examination was requested in 60%, 87%, and 3% of visits to private GPs, specialists, and CHCs, respectively. Three out of four private GPs who requested a sputum examination also requested a CXR in contrast to one (3%) GP at the CHC who requested a CXR in addition to a sputum test. Antibiotics were prescribed in 77% of visits to private GPs, compared to 53% of visits to specialists and 30% to CHCs. These excluded prescriptions of anti-TB drugs reported in 4% of private GP visits and 20% of visits to specialists. One-third of the TB-drug prescriptions made by specialists were also accompanied by prescriptions for other antibiotics. Prescription of corticosteroids were reported in up to 31% of the visits.

For Scenario B, where a request for CXR examination was expected, adherence to NTP guidelines by private GPs (n = 60) and specialists (n = 18) were 88% and 89%, respectively (Fig 3B and S1 Table). Repeat sputum examination was requested in 12% of private GP visits and 11% of specialist visits. Antibiotics were prescribed in 58% of visits to private GPs and 39% of visits to specialists. In addition, anti-TB drugs were prescribed by 8% of the private GPs and 28% of specialists, whereas corticosteroids were prescribed in 28% and 11% of the visits, respectively.

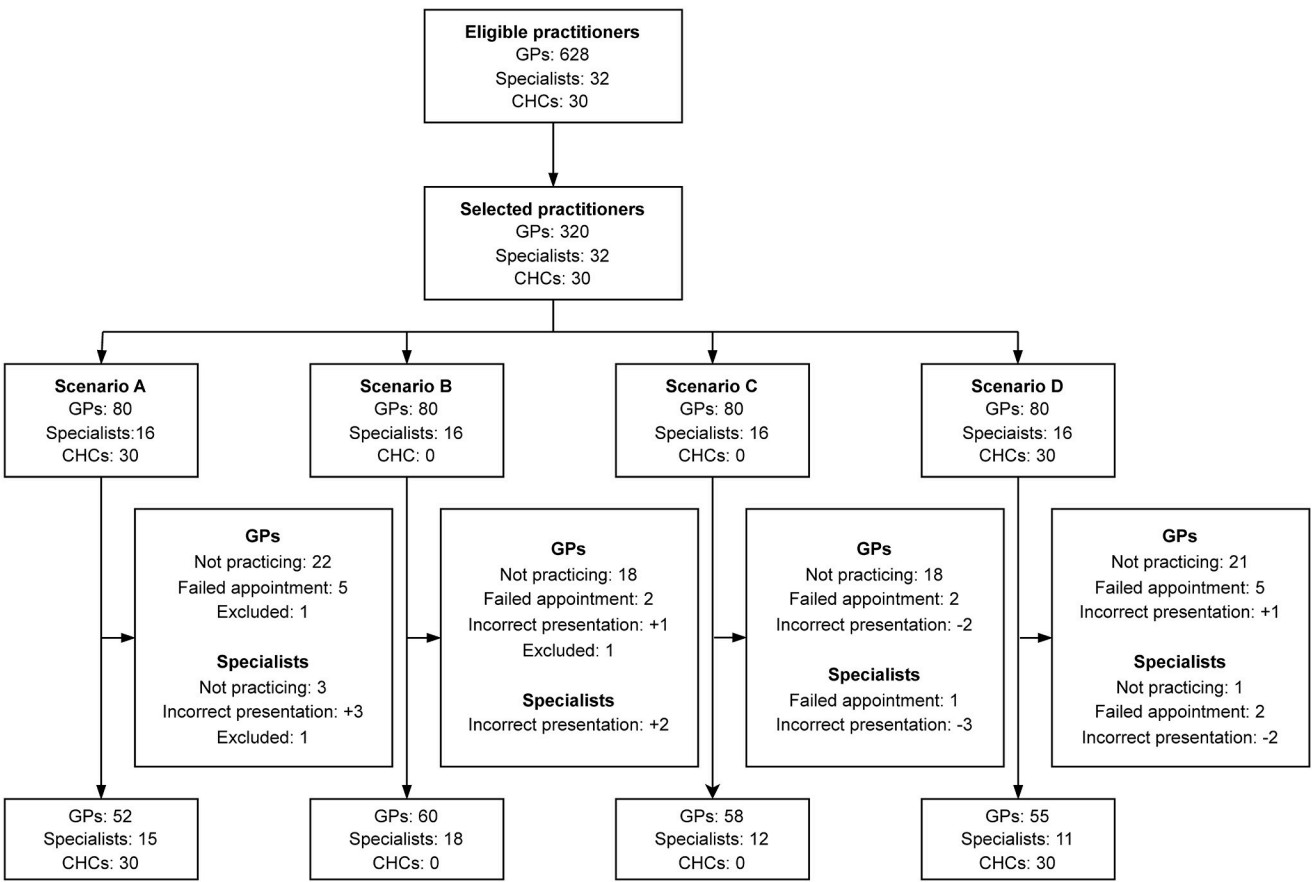

**Fig 2. Implementation of standardized patient visits to providers.** Private General Practitioners (GPs) received one standardized patient (SP) visit while specialists received two visits of different scenarios. Incorrect presentations led to scenario switches. Visits that resulted from a physician being visited more times (e.g., due to switching practice time) than planned were excluded from analysis.

For Scenario C, the recommended TB drug regimen was prescribed in 12% of the visits to private GPs (n = 58), compared to 25% of the specialists (n = 12) (Fig 3C and S1 Table). When referral for treatment was considered as a correct management for the presented scenario, adherence to NTP guidelines were 81% and 67% for private GPs and specialists, respectively. Repeat sputum examination was requested by 10% and 8% of private GPs and specialists, respectively, whereas CXR examination was requested by 59% of private GPs and 83% of specialists. Broad spectrum antibiotics were prescribed by 21% of private GPs and 17% of specialists. In addition, in 7% of the visits to private GPs, corticosteroids were prescribed to the SPs.

For Scenario D, adherence to NTP guidelines to refer cases for Xpert examination among private GPs (n = 55), specialists (n = 11), and CHCs (n = 30) was 4%, none, and 3%, respectively (Fig 3D; S1 Table). One private GP (2%) ordered a bacteriological culture. In addition, 44% of private GPs, 10% of specialists, and 90% of CHCs ordered a sputum smear examination. Requests for CXR were made by 73% of private GPs and 91% of specialists, compared to 10% from the CHCs. Anti-TB drugs were prescribed in 11% of the visits to private GPs and 18% to specialists, compared to none of the visits to the CHCs. Additionally, broad spectrum antibiotics were prescribed by 60%, 36%, and 7% of private GPs, specialists, and CHCs, respectively. None of the specialists prescribed corticosteroids, compared to 20% of private GPs and 7% of CHCs.

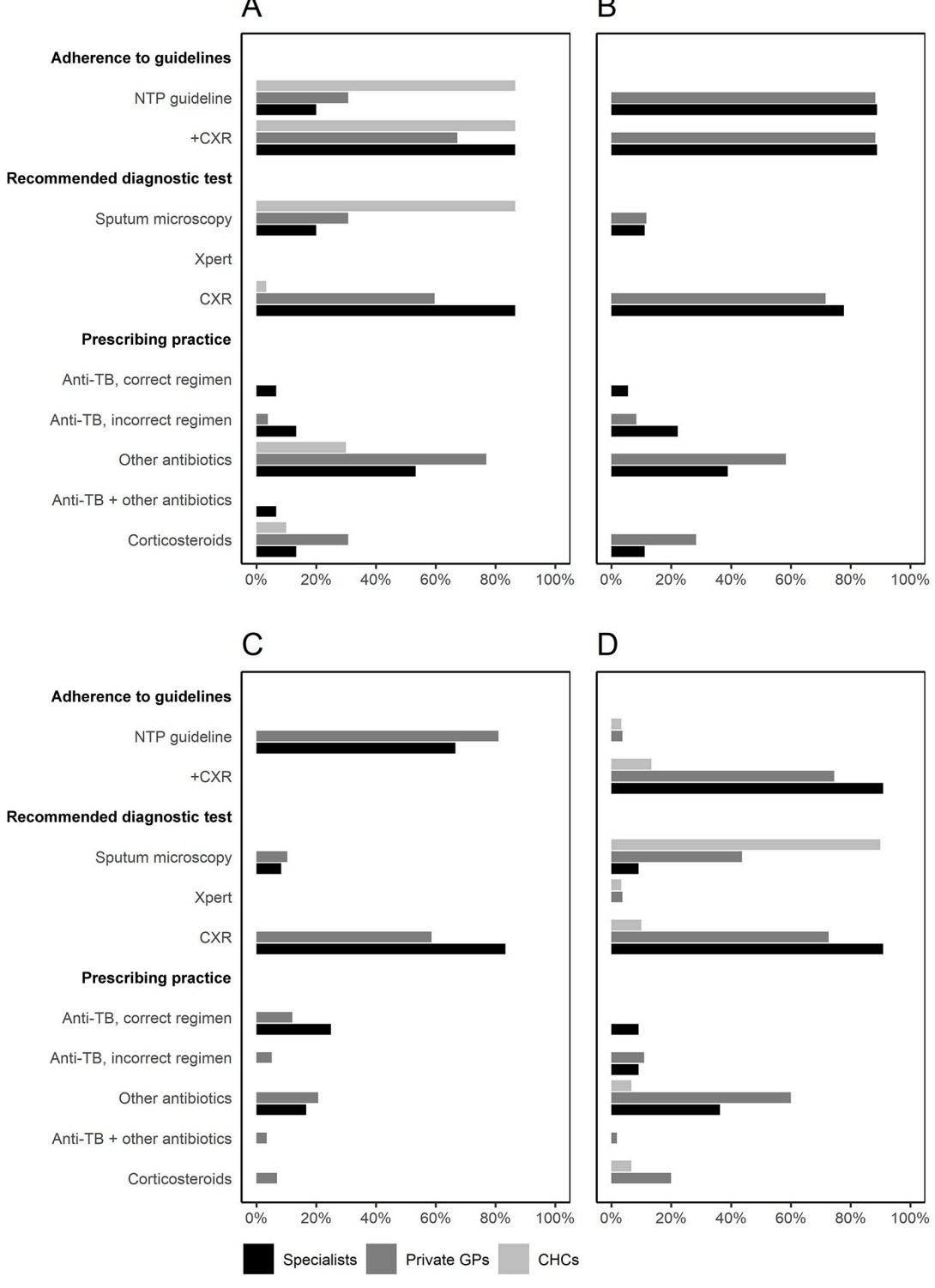

**Fig 3. Main outcomes, expressed as percentages according to clinical case scenario and the types of providers.** A) Scenario A: Classic case of presumptive TB; no sputum smear result; no previous history of TB; B) Scenario B: Classic case of presumptive TB; a recent negative sputum smear result; no previous history of TB; C) Scenario C: Classic case of presumptive TB; a recent positive sputum smear result; no previous history of TB; D) Scenario D: Classic case of presumptive TB; no sputum smear result; a history of incomplete TB treatment. Abbreviations: CHCs: Community Health Centers, GPs: General Practitioners, CXR: chest x-ray. Scenarios B and C were not presented to CHCs. +CXR indicates adding CXR request as a correct case management for scenarios A, B, and D.

### Other physician-SP interactions

Clinical consultation with both private GPs and specialists lasted for a median of 10 minutes (IQR 7–15) compared to 8 minutes (IQR 5–11.5) at the CHCs. During these brief interactions, GPs at the CHCs tended to focus more on classical TB symptoms (e.g., presence of fever, night sweat, weight loss, and history of TB) compared to PPs (Fig 4 and S2 Table). However, specialists seemed to perform more thorough clinical assessments. History of comorbidities, such as diabetes and HIV, were rarely asked by any providers.

Providers communicated their diagnoses with the SPs in most of the interactions. The most often communicated diagnosis was TB (S3 Table). In Scenarios A and D, GPs at CHCs more frequently communicated TB diagnosis (64% and 90% for Scenario A and D, respectively) compared to PPs. Other commonly communicated diagnoses included bronchitis (up to 16% by private GPs seeing Scenario A) and upper respiratory tract infection. These two diagnoses were also communicated by a small number (3%) of private GPs presented with Scenario C, despite their positive sputum smear result. On the other hand, specialists were the least likely among the providers to share any diagnosis with the SPs (up to 29% in Scenario D).

Referrals of SPs to other health facilities for follow-up diagnosis workups and/or treatment were common, especially from PPs. Out of 60 visits to CHC, referrals were done for CXR (1%), sputum examination (20%), and treatment (23%). By contrast, out of the 225 visits to private GPs, referrals for CXR, sputum examination, and treatment were made in 60%, 22%, and 42% of the visits (not mutually exclusive), respectively. Referral percentages for CXR, sputum examination and treatment by specialists were 75%, 11%, and 18% (not mutually exclusive), respectively. Public facilities were preferred by all providers for treatment referral, whereas private facilities were preferred by PPs for CXR and sputum examinations, especially by specialists. Clinical consultation with PPs costed an average of USD 6.60 (2018 exchange rate), compared to USD 0.2 for visits to CHC.

## Discussion

We used SPs to evaluate the adherence of PPs to the NTP guidelines in managing TB cases in an urban setting in Indonesia. For adult patients presenting with complaints suggestive of classical pulmonary TB, less than a third of the PPs managed the simulated cases in accordance with NTP guidelines. By contrast, we observed over 80% adherence among practitioners in the CHCs. The lack of adherence by PPs can be explained, at least partially, by their clear preference for using CXR, as opposed to sputum testing, as a triage procedure. With respect to treatment, PPs often prescribed broad spectrum antibiotics to patients presenting with classical TB complaints. In addition, in up to a quarter of the scenario visits, PPs prescribed anti-TB drugs that were not in line with guideline recommendations. Considering that many TB patients seek care from PPs, our findings suggest that TB case management by PPs is suboptimal, and that there is a need to further engage them to improve TB control in Indonesia.

The clear preference of PPs for recommending CXR over microbiological testing, especially among those with higher qualifications, is consistent with findings from studies using a similar approach in China and India [7, 8]. When used as an initial test and followed up with a microscopy or Xpert testing, CXR can increase triage sensitivity in detecting TB cases [15] and may reduce delay to treatment [16]. However, the use of CXR can lead to under- or over-diagnosis of pulmonary TB unless it used together with bacteriological testing in the triage of symptomatic individuals. Our study did not explore the reasoning behind physicians' preferences for CXR and whether they would have ordered a follow up bacteriological confirmatory test. Nevertheless, it is likely that many PPs provide TB treatment without attempting bacteriological

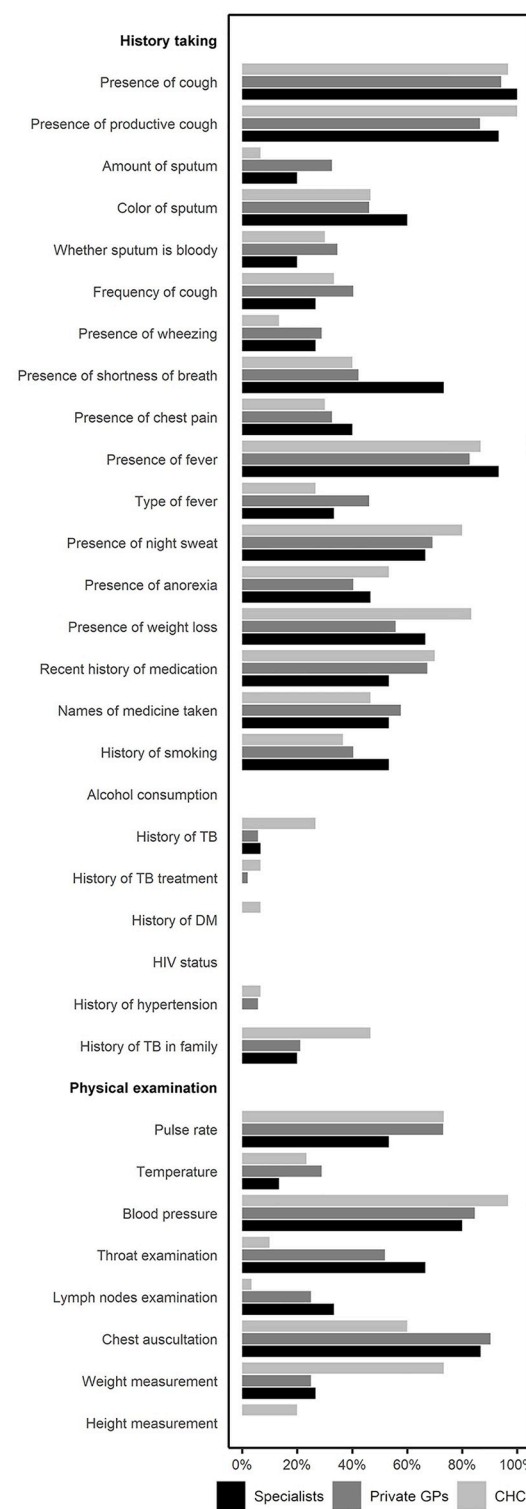

**Fig 4. Clinical history taking and examination by different providers in Scenario A.** Scenario A: Patients presenting with complaints indicative of pulmonary tuberculosis. Abbreviations: CHCs: Community Health Centers, GPs: General Practitioners.

confirmation. Studies from India [17], and Pakistan [18] showed PPs' compliance to be low (under 60%) for the utilization of sputum microscopy in TB diagnosis.

Following clinical examination, most physicians communicated the possible diagnosis to the SPs and the likelihood of suggesting a TB diagnosis was higher when they were presented with more conclusive evidence of disease (i.e., positive sputum smear result). Nonetheless, different diagnoses were also communicated to the SPs with bronchitis being the most common, but we did not explore the rationale behind these other diagnoses. Lastly, prescription of broad-spectrum antibiotics or a sub-standard TB regimen by PPs was not uncommon, even for bacteriologically confirmed TB patients. In some instances, the correct TB regimen was prescribed in the absence of supporting indications. Inappropriate use of antibiotics for presumptive TB patients were also reported by similar studies from India, China, and Kenya [19]. The findings in relation to diagnosis and treatment by PPs may suggest lack of knowledge among PPs about the TB management guidelines as has been shown elsewhere in Indonesia [20], compared to their counterparts in the public sector who were shown to have good knowledge about TB [21]. Alternatively, it may be the case that a 'know-do gap' exists among PPs [7]. Indeed, a previous study in Indonesia showed the majority of surveyed PPs reported that they would order a sputum test for presumptive TB patients [4]. Likely explanations for this lack of knowledge and know-do gap include lack of exposure to the TB program [22], and lack of access to diagnostic facilities [3], although not in terms of geographic distance for an urban area like Bandung. Other perceived barriers to referral, such as opportunity cost may be important [23].

The underutilization of Xpert in cases suspected of drug-resistant TB, particularly by physicians at the CHC, may indicate the prevailing understanding of the use of Xpert and/or the perceived lack of access to the diagnostic modality. At the time of the study, Xpert had only recently been introduced into the NTP guidelines, and in Bandung only five hospitals were equipped with the machine, and no CHC had one. A perceived lack of access or difficulties in accessing the modality therefore likely resulted from the limited availability of Xpert. On the other hand, lack of knowledge about Xpert seemed common among PPs [24, 25].

Methodologically, this study demonstrates the use of SPs is beneficial in evaluating the quality of TB care in PPs against the prevailing national guidelines. Our study found that twice as many private GPs did not use sputum examination for diagnosing presumptive adult pulmonary TB cases than a previous survey [4], possibly confirming the superiority of SP method to overcome social desirability bias. Our study also demonstrated the importance of using professional and experienced actors as SPs who are "able to think on their feet" in making the observed experience as natural as possible [13].

Our study does have some limitations. First, the method is resource intensive thereby limiting the sample size and its scope of evaluation. The scenarios were developed to represent adult pulmonary TB cases and therefore did not include other forms of TB (e.g., latent TB or drug-resistant TB) or TB in other population groups (e.g., in children or in people living with comorbidities). The quality of service for adult pulmonary TB cases, however, can serve as best-case scenario considering the added complexities in the management of other forms of TB or TB in other population subgroups. Further, our sample was drawn from one urban health care area and therefore it lacks representativeness for a country like Indonesia. However, considering that Bandung is an urban area with sufficient geographical access to TB-related care, we believe that our findings represent the healthcare system context from a part of Indonesia considered to be better-off and, therefore, other places in Indonesia may have similar or worse TB care in the private sector. The small sample size prevented us to conduct further subgroup analysis or to identify behaviors with very rare occurrences. However, the sample size was adequate to answer our research question and it has been argued that small

sample size does not necessarily reduce SP study generalizability because each observation could be seen as a valid snapshot of the service experience [13]. We only evaluated one-time doctor-patient interactions and hence our findings may not reflect adherence to NTP guidelines over the whole course of doctor-patient interactions.

Many TB control-relevant contexts have changed since the study was conducted, including national and international TB guidelines. However, we believe that the general findings and key takeaway messages continue to be important when considering the role of private practitioners (PPs) in TB control. Our study highlighted and strengthened findings from previous studies on the lack of adherence of PPs to the prevailing guidelines. Other studies have attempted to explore the reasons for this incongruence and pointed to the lack of awareness about the prevailing guidelines [26] or barriers to following the guidelines, such as lack of access to diagnostic facilities [27]. Indeed, the rapid development of technologies for TB (e.g., diagnostics and drugs, case finding strategies) followed by relatively rapid adoption into policy potentially widen the knowledge and know-do gaps among PPs.

## Conclusion

Our study highlights the usefulness of using SPs to assess PPs' practice against TB guidelines in Indonesia. It identifies areas where the roles of PPs in TB control could be enhanced, including education about standard diagnostics such as sputum smear microscopy or Xpert MTB/RIF test, or other tests as new guidelines are being issued. Enhancing the understanding of the role of CXR in TB diagnosis may also strengthen TB case management among PPs. Carefully designed SPs studies could be valuable for evaluating the success of interventions aimed at increasing PPs' adherence to TB control guidelines and assessing the quality of TB care in general. They are particularly beneficial in identifying and addressing discrepancies between what practitioners know and what they do using direct observation of real-world practice.

## Supporting information

**S1 Table. Main outcomes of standardized patient visits to private and public practitioners, according to clinical case scenario and the types of providers.**
(DOCX)

**S2 Table. Clinical enquiries, expressed as percentages and 95% confidence intervals (CIs), performed by different providers under different patient scenarios.**
(DOCX)

**S3 Table. Diagnosis communicated to standardized patients (SPs) by providers according to scenario.**
(DOCX)

## Acknowledgments

We would like to thank the Bandung City Health Office for their supports, the dedicated research assistants at INSTEP study for their invaluable contributions, and the talented team of actors and actresses who faithfully presented the cases.

## Author Contributions

**Conceptualization:** Bony Wiem Lestari, Panji F. Hadisoemarto, Susan McAllister, Reinout van Crevel, Philip C. Hill, Bachti Alisjahbana.

**Data curation:** Nur Afifah, Deny Fattah.

**Formal analysis:** Bony Wiem Lestari, Panji F. Hadisoemarto, Argita D. Salindri, Bachti Alisjahbana.

**Funding acquisition:** Philip C. Hill, Bachti Alisjahbana.

**Investigation:** Bony Wiem Lestari, Panji F. Hadisoemarto, Nur Afifah, Deny Fattah.

**Methodology:** Bony Wiem Lestari, Panji F. Hadisoemarto, Susan McAllister, Reinout van Crevel, Philip C. Hill, Bachti Alisjahbana.

**Project administration:** Nur Afifah, Deny Fattah.

**Resources:** Bachti Alisjahbana.

**Supervision:** Susan McAllister, Reinout van Crevel, Megan Murray, Philip C. Hill, Bachti Alisjahbana.

**Validation:** Bony Wiem Lestari, Panji F. Hadisoemarto, Nur Afifah, Deny Fattah.

**Visualization:** Panji F. Hadisoemarto, Nur Afifah, Deny Fattah.

**Writing – original draft:** Bony Wiem Lestari, Panji F. Hadisoemarto.

**Writing – review & editing:** Bony Wiem Lestari, Panji F. Hadisoemarto, Nur Afifah, Susan McAllister, Deny Fattah, Argita D. Salindri, Reinout van Crevel, Megan Murray, Philip C. Hill, Bachti Alisjahbana.

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
