## [Decision Letter · Decision Letter 0]

7 Dec 2023

PGPH-D-23-02033

Tuberculosis care provided by private practitioners in an urban setting in Indonesia: findings from a standardized patient study

Dear Dr. Hadisoemarto,

Thank you for submitting your manuscript to PLOS Global Public Health. After careful consideration, we feel that it has merit but does not fully meet PLOS Global Public Health’s publication criteria as it currently stands. Therefore, we invite you to submit a revised version of the manuscript that addresses the points raised during the review process.

I would like to sincerely apologise for the delay you have incurred with your submission. It has been exceptionally difficult to secure reviewers to evaluate your study. We have now received two completed reviews; the comments are available below. The reviewers have raised significant scientific concerns about the study that need to be addressed in a revision.

Please revise the manuscript to address all the reviewer's comments in a point-by-point response in order to ensure it is meeting the journal's publication criteria. Please note that the revised manuscript will need to undergo further review, we thus cannot at this point anticipate the outcome of the evaluation process.

We look forward to receiving your revised manuscript.

Kind regards,

Miquel Vall-llosera Camps

Staff Editor

Journal Requirements:

1. We do not publish any copyright or trademark symbols that usually accompany proprietary names, eg  ©, ®, ™  (e.g. next to drug or reagent names). Please remove all instances of trademark/copyright symbols throughout the text, including ® on page 4.

2. Please amend your Data Availability Statement and indicate where the data may be fou

Reviewers' comments:

Reviewer's Responses to Questions

**Comments to the Author**

1. Does this manuscript meet PLOS Global Public Health’s publication criteria? Is the manuscript technically sound, and do the data support the conclusions? The manuscript must describe methodologically and ethically rigorous research with conclusions that are appropriately drawn based on the data presented.

Reviewer #1: Partly

Reviewer #2: Partly

2. Has the statistical analysis been performed appropriately and rigorously?

Reviewer #1: No

Reviewer #2: Yes

3. Have the authors made all data underlying the findings in their manuscript fully available (please refer to the Data Availability Statement at the start of the manuscript PDF file)?

Reviewer #1: Yes

Reviewer #2: No

4. Is the manuscript presented in an intelligible fashion and written in standard English?

Reviewer #1: No

Reviewer #2: Yes

5. Review Comments to the Author

Reviewer #1: The topic on TB care in private health care setting is really interesting and the methodology is novel. However, some Major concerns in the study are:

The study was conducted in Bandung city from July 2018 to April 2019 - which is almost before 4 years - so all the guidelines related to TB care has been changed a lot globally. For instance, one of the finding like preference of Chest Xray over sputum examination for TB diagnosis does not remain important - because as per latest guidelines, sputum examination and Chest X-ray are recommended simultaneously for TB diagnosis.

The sample size has been calculated based on Indian study.

The four case scenarios mentioned in the article does not include all situations of TB infection.

All the figures / flow charts are cluttered and not clear.

The manuscript is suitable for national journal.

Reviewer #2: This manuscript shows originality and demonstrates good knowledge about the topic. The discussion section is well articulated and relates directly to the results of the study's result and no doubt will add to existing knowledge in the respective field. In addition, the research meets all applicable standards for the ethics of experimentation and research integrity.

However, there are few areas that needed to be updated to make the manuscript ready for submission:

1. The study's goal and objectives are not clearly stated. I personally struggled to understand what the study aimed to achieve - Stating clearly the study's aim and objective will be beneficial to readers and will improve its quality.

2. The method was discussed in details; however, readers are left to figure out the exact methodology used in this study - I suggest you clearly state your methods in addition to providing details to enhance reproducibility. Also, the method and tolls for analysis is not well stated.

3. It does not fully meets the journal's criteria; for instance, there is no section on the conclusion, and the data used for the analysis was not made available - Extracting the conclusion part from the discussion and ensuring that it speaks to the result of the study will be very helpful.

4. Few typos and grammatical errors were observed throughout the sections - There is the need to conduct multiple proof reading before resubmission.

Conclusion

This manuscript has good potential and stands a good chance to be published in this journal but needs some revision as already suggested.

6. PLOS authors have the option to publish the peer review history of their article (what does this mean?). If published, this will include your full peer review and any attached files.

**Do you want your identity to be public for this peer review?** For information about this choice, including consent withdrawal, please see our Privacy Policy.

Reviewer #1: **Yes: **Kedar Mehta

Reviewer #2: **Yes: **Joseph Kuye

---

## [Decision Letter · Decision Letter 1]

15 May 2024

Tuberculosis care provided by private practitioners in an urban setting in Indonesia: findings from a standardized patient study

PGPH-D-23-02033R1

Dear Dr Hadisoemarto,

We are pleased to inform you that your manuscript 'Tuberculosis care provided by private practitioners in an urban setting in Indonesia: findings from a standardized patient study' has been provisionally accepted for publication in PLOS Global Public Health.

Best regards,

Julia Robinson

Executive Editor

Reviewer Comments (if any, and for reference):

Reviewer's Responses to Questions

**Comments to the Author**

1. If the authors have adequately addressed your comments raised in a previous round of review and you feel that this manuscript is now acceptable for publication, you may indicate that here to bypass the “Comments to the Author” section, enter your conflict of interest statement in the “Confidential to Editor” section, and submit your "Accept" recommendation.

Reviewer #2: All comments have been addressed

2. Does this manuscript meet PLOS Global Public Health’s publication criteria? Is the manuscript technically sound, and do the data support the conclusions? The manuscript must describe methodologically and ethically rigorous research with conclusions that are appropriately drawn based on the data presented.

Reviewer #2: Yes

3. Has the statistical analysis been performed appropriately and rigorously?

Reviewer #2: Yes

4. Have the authors made all data underlying the findings in their manuscript fully available (please refer to the Data Availability Statement at the start of the manuscript PDF file)?

Reviewer #2: Yes

5. Is the manuscript presented in an intelligible fashion and written in standard English?

Reviewer #2: Yes

6. Review Comments to the Author

Reviewer #2: The author responded sufficiently to the recommendations from the review, which is satisfactory. The overall quality of the manuscript has improved and is ready for publication.

My overall recommendation is that the paper be accepted and published.

Regards

7. PLOS authors have the option to publish the peer review history of their article (what does this mean?). If published, this will include your full peer review and any attached files.

**Do you want your identity to be public for this peer review?** For information about this choice, including consent withdrawal, please see our Privacy Policy.

Reviewer #2: **Yes: **JOSEPH KUYE
